# TICKET-BENCH: A KICKOFF FOR MULTILINGUAL AND REGIONALIZED AGENT EVALUATION

## ABSTRACT

Large language models (LLMs) are increasingly deployed as task-oriented agents, where success depends on their ability to generate accurate function calls under realistic, multilingual conditions. However, existing evaluations largely overlook cultural and linguistic diversity, relying on monolingual or naïvely translated datasets. We introduce Ticket-Bench, a benchmark for multilingual function-calling in task-oriented scenarios. Ticket-Bench simulates the domain of soccer ticket purchases across six major languages—Portuguese, English, Spanish, German, Italian, and French—using localized teams, cities, and user profiles to ensure cultural authenticity. We evaluate a wide range of commercial and open-source LLMs, measuring function-calling accuracy and consistency across languages. Results show that reasoning-oriented models (e.g., GPT-5, Qwen3-235B) dominate performance but still exhibit notable cross-lingual disparities. These findings underscore the need for culturally aware, multilingual benchmarks to guide the development of robust LLM agents.

## 1 INTRODUCTION

Large language models (LLMs) have quickly evolved from mere text generators to agents capable of orchestrating real-world actions through function-calling and tool use Patil et al. (2024); Schick et al. (2023). This paradigm shift has fueled the adoption of LLMs in a wide array of digital assistants and task automation platforms, where interpreting user requests and triggering appropriate actions is essential Guo et al. (2024a); Li et al. (2024).

A critical gap in current research is the absence of multilingual, culturally aware benchmarks for evaluating function-calling. Existing evaluations of tool use and agent performance cover important ground but are predominantly English-centric Mohammadi et al. (2025); Patil et al. (2025); Castillo-Bolado et al. (2024); Barres et al. (2025). Related efforts on general task completion and information retrieval extend to multiple languages Huang et al. (2025); Chirkova et al. (2024), yet they often depend on monolingual or simply translated datasets. In real deployments, users converse with assistants in many languages and reference region-specific entities that shape model interactions and may influence how well a model executes function calls. Without benchmarks that reflect this linguistic and cultural localization, we cannot reliably assess—or improve—models' ability to plan and fulfill real-world tasks across different regions.

To address this gap, we introduce Ticket-Bench, a benchmark for evaluating LLM function-calling in the domain of purchasing soccer game tickets. Ticket-Bench features tasks in six major languages–Portuguese, English, Spanish, German, Italian, and French. We carefully localize user queries and context, adapting city names, team names, and contextual nuances to each language and region. This approach ensures that LLMs are tested not only for multilingual understanding but also for their ability to handle realistic scenarios. Ticket-Bench is available at `https://anonymous.4open.science/r/Ticket-Bench-5BC0`.

Ticket-Bench provides a wide range of scenarios, requiring LLMs to interpret nuanced constraints and user preferences when generating structured function calls to interact with the system. Our evaluations reveal challenges in some LLMs to interpret user intent and produce the expected actions robustly across all tested languages.

The main contributions of our paper are as follows:

- We introduce Ticket-Bench, a benchmark designed to evaluate LLM agent capabilities in ticket-purchasing scenarios, featuring over 1000 evaluation cases across six languages with contextually adapted environments.

- Ticket-bench provides an LLM-free, programmatic evaluation that checks the final environment state (expected tickets booked, no unexpected bookings). To capture robustness, we report a pass^3 consistency metric computed over multiple executions per query, rewarding models that solve tasks reliably, not just once.

- We observe systematic, family-specific language asymmetries: no language is uniformly "easy" or "hard," but some model families show notable strengths and deficits across certain languages, likely due to language imbalances in the model training data.

## 2 RELATED WORK

**Multilingual disparities in LLMs.** Recent studies have shown that LLMs perform unevenly across regions, particularly in underrepresented countries and cultural contexts. For example, World-Bench Moayeri et al. (2024) exposes gaps in factual recall tied to economic and geographic divides, TiEBe Almeida et al. (2025) highlights inconsistencies in capturing temporally grounded events, and BLEnD Myung et al. (2024) uncovers cultural and linguistic biases in everyday knowledge.

Towards the multilingual and culturally grounded evaluation, Multi-IF He et al. (2024) introduces a benchmark focused on multi-turn and multilingual instruction-following, exploring whether models can maintain coherence and correctly execute instructions across different languages over multiple dialogue turns. However, its scope remains restricted to textual instruction comprehension.

While multilingual disparities are explored in text-based tasks, it is unclear how they impact LLMs function-calling capabilities, where models must correctly interpret and execute structured calls across languages and contexts.

**English-Only Function Call Benchmarks.** Most function-calling benchmarks for LLMs emphasize interaction realism and dialogue robustness, simulating policy-constrained dialogues or API-driven tasks. However, these evaluations are largely monolingual and culturally neutral, leaving open how models adapt to multilingual, localized settings.

$\tau^2$**-Bench** Barres et al. (2025) extends the original $\tau$-Bench Yao et al. (2024) with dual-control environments, where both agent and simulated user act on a shared state, and provides analyses that separate reasoning from coordination errors. **ConFETTI** Alkhouli et al. (2025) evaluates turn-level function-calling across 109 human-simulated conversations (313 user turns) and 86 APIs, testing goal changes, follow-ups, and chained calls. Both benchmarks focus on turn-level function-calling, assessing how well LLMs manage multi-turn interactions, handle dynamic goals, and coordinate with simulated users in English-only settings.

**HammerBench** Wang et al. (2025) evaluates LLM tool usage in long-context mobile assistant scenarios, incorporating multi-step task execution, error recovery, and realistic API sequences. **Stable-ToolBench** Guo et al. (2024b) focuses on robustness and reliability across diverse APIs, emphasizing consistency in multi-step interactions and using a virtual API server for stable evaluation. These benchmarks assess interactive, multi-step tool usage, testing how LLMs maintain context, execute workflows, and recover from errors.

**BigCodeBench** Zhuo et al. (2024) evaluates LLM code generation across multiple programming languages and frameworks. **BFCL** Patil et al. (2025) focuses on chain-of-thought reasoning and problem-solving without interactive environment control. **API-Bank** Li et al. (2023) provides a collection of APIs and tasks for evaluation, focusing on single-task multi-step API usage; it does not involve multi-turn dialogue or agent-like decision making. These benchmarks share a focus on evaluating LLMs' function-calling capabilities in single-task, multi-step scenarios, reasoning, and other aspects, but remain limited to only English.

**Multilingual Function-Calling Studies.** Recent work has begun to explore how multilingual contexts affect LLMs' function-calling capabilities. For instance, **BenchMAX** Huang et al. (2025) introduced a multilingual evaluation suite with a Tool Use track, assessing models' ability to invoke correct functions across multiple languages through simple translation of the nexus team (2023)

dataset. Similarly, **ACEBench** Chen et al. (2025) extends function-calling evaluation to both English and Chinese, providing insights into cross-lingual performance.

However, existing multilingual function-calling efforts remain limited in scope and experimental control. Most cover only a small set of languages or rely on direct translations, and do not cover the localization of relevant entities during evaluation. Furthermore, many evaluations are dependent on LLM-as-judge or turn-level signals instead of verifiable end-state outcomes.

**TicketBench** provides a simulated, culturally grounded environment across six languages, with synchronized schedules, localized user profiles, and aligned question templates to ensure comparability. Models interact through a fixed set of fully translated functions with standardized inputs and outputs, allowing evaluation of reasoning and execution in multi-step function calls while isolating language-specific ambiguities. This design enables a more systematic and robust multilingual assessment than prior benchmarks.

Table 1 summarizes the key characteristics of the benchmarks reviewed alongside TicketBench. *Languages* indicates the number of languages supported by the benchmark; *Regional Adaptation* reflects whether datasets incorporate localized attributes or cultural context; *Interactiveness* denotes whether the model's outputs dynamically influence the environment; *Multi-Step* specifies whether tasks require sequential or dependent operations to fulfill the task; *LLM-Free Evaluation* indicates whether correctness can be assessed without relying on another LLM as a judge; and *System Focus* identifies benchmarks designed to evaluate full agent pipelines or realistic system workflows rather than isolated function calls.

| Benchmark | Languages | Regional Adaptation | Interactiveness | Multi-Step | LLM-Free Evaluation | System Focus |
|---|---|---|---|---|---|---|
| $\tau^2$-Bench | 1 | × | ✓ | × | ✓ | ✓ |
| ConFETTI | 1 | × | ✓ | × | × | × |
| HammerBench | 1 | × | × | ✓ | × | × |
| StableToolBench | 1 | × | × | ✓ | × | × |
| BigCodeBench | 1 | × | × | ✓ | ✓ | × |
| BFCL | 1 | × | × | ✓ | ✓ | × |
| API-Bank | 1 | × | × | ✓ | ✓ | × |
| ACEBench | 2 | × | ✓ | ✓ | ✓ | ✓ |
| BenchMAX | 17 | × | × | ✓ | ✓ | × |
| **Ticket-Bench** | **6** | ✓ | ✓ | ✓ | ✓ | ✓ |

Table 1: Comparison of function-calling benchmarks, highlighting multilingual coverage, regional adaptation, interactiveness, multi-step evaluation, LLM-free assessment, and system focus.

## 3 METHODOLOGY

### 3.1 ENVIRONMENT AND ENTITIES

To evaluate function-calling capabilities in multilingual scenarios, we constructed a simulated ticket-purchasing environment with three main components.

**Users** are defined by a culturally appropriate name (sampled from common names in the target country), a virtual account balance, and a preferred soccer team. These attributes introduce personal constraints–such as affordability and team preference—into the simulation. For each language, we generate 20 users, ensuring that no two share the same preferred team.

**Game schedules** form the core set of events. Each schedule represents a full league season, with games specifying the home team, away team, city, stadium, ticket price, and date. All entity names (teams, cities, stadiums) are localized to the target language and region. We simulate two types of schedules per language: one where each of the 20 teams plays every other team once, and another where they play twice, producing a total of 380 matches—the same number found in most professional leagues that inspired our setup.

**Leaderboards** capture historical league performance, enabling queries that depend on past results. For each season, they record statistics for each team, including points, wins, draws, losses, goals scored, and goals conceded. To generate these tables, we synthetically assign goals to each match while enforcing consistency, ensuring the resulting distributions resemble realistic league outcomes.

Table 2: Constraint coverage across the 17 templates. A checkmark (✓) indicates that the constraint is present in the template.

| ID | Template (abridged) | Semester | Weekday | Price | Location | Leaderboard |
|----|---------------------|----------|---------|-------|----------|-------------|
| 1 | Next {user_team} game I can afford | | | | | |
| 2 | Next game of my team I can afford | | | | | |
| 3 | Next game I can afford, first semester | ✓ | | | | |
| 4 | Next game I can afford, not on weekend | | ✓ | | | |
| 5 | Cheapest game this year | | | ✓ | | |
| 6 | Next game in {location} | | | | ✓ | |
| 7 | Next game vs team with >60 points in {year} | | | | | ✓ |
| 8 | Next game, second semester, midweek | ✓ | ✓ | | | |
| 9 | Most expensive game I can afford, not weekend | | ✓ | ✓ | | |
| 10 | Cheapest game in {location} | | | ✓ | ✓ | |
| 11 | Next game in {location}, vs top 8 teams of {year} | | | | ✓ | ✓ |
| 12 | Cheapest game, second semester, midweek | ✓ | ✓ | ✓ | | |
| 13 | Most expensive game I can afford, not weekend, in {location} | | ✓ | ✓ | ✓ | |
| 14 | Cheapest game in {location}, vs team with >20 goals {year} | | | ✓ | ✓ | ✓ |
| 15 | Most expensive game I can afford, 2nd semester, midweek, in {location} | ✓ | ✓ | ✓ | ✓ | |
| 16 | Cheapest game I can afford, not weekend, in {location}, vs >20 goals {year} | | ✓ | ✓ | ✓ | ✓ |
| 17 | Most expensive game, 2nd semester, not weekend, in {location}, vs top 3 of {year1}/{year2} | ✓ | ✓ | ✓ | ✓ | ✓ |

## 3.2 QUESTION TEMPLATES

We define 17 question templates representing distinct ticket-purchasing scenarios. Each template combines constraints drawn from five categories:

- **Semester:** restricts the date to the first or second semester.
- **Weekday:** restricts the day of the week (e.g., avoids weekends or selects midweek days).
- **Price:** selects the cheapest or most expensive game that satisfies other constraints.
- **Location:** restricts the game to a specific city.
- **Leaderboard:** adds conditions based on past results (e.g., top $n$ teams, teams with more than $x$ goals or points).

Table 2 summarizes the 17 templates and the constraints each one involves. The actual templates can be found in the Appendix B.

For each question template, we instantiate ten unique queries by varying user profiles, game schedules, and constraint specifications. This results in 170 queries per language and a total of 1,020 queries across the six languages in TicketBench.

To capture a wider range of scenarios, we design a subset of queries for which no valid booking exists. Specifically, 15% of the questions are constructed so that no game in the schedule satisfies the stated constraints. This guarantees that models are not only evaluated on their ability to find valid matches, but also on their capacity to correctly detect when no solution exists.

## 3.3 AVAILABLE FUNCTIONS

Models are provided with a fixed set of five callable functions to solve the user query. Each function exposes a simple interface with clearly defined inputs and outputs, ensuring consistency across languages.

**Get User Info** retrieves the user profile, including the user's name, preferred team, and current account balance. This function allows models to check affordability constraints and align ticket choices with the user preferences.

**List Games** returns a paginated list of games, with a maximum of 10 entries per page. The function accepts optional filters on fields such as location and team, and also supports ordering (e.g., by date or by price). Each game is represented with its identifier, teams, city, stadium, ticket price, and scheduled date. This function is central for exploring the search space of possible tickets.

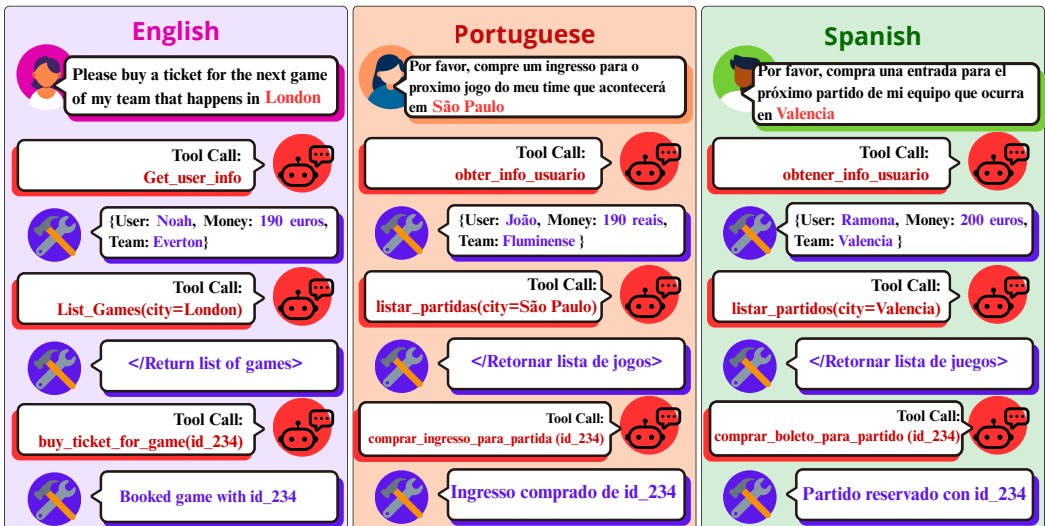

Figure 1: Example of Ticket Bench question localization

**Buy Game Ticket** finalizes the decision process by purchasing a ticket for a given game identifier. Successful execution updates the environment state by reducing the user's balance and marking the corresponding game as booked.

**Get Leaderboard** provides access to historical league performance. It returns, for a specified year, a table of per-team statistics including points, wins, draws, losses, goals scored, and goals conceded. This function enables queries that depend on conditions such as "top $n$ teams" or "teams with more than $x$ goals."

**Get Weekday from Date** returns the day of the week corresponding to a given date string in the format YYYY-MM-DD. This function supports constraints involving weekends or midweek games.

## 3.4 LOCALIZATION

We instantiate the environment in six languages, each aligned with a major national soccer league: Brasileirão in Portuguese (Brazil), Ligue 1 in French (France), Bundesliga in German (Germany), La Liga in Spanish (Spain), Serie A in Italian (Italy), and Premier League in English (United Kingdom), more information about our selection of Leagues can be found in the Appendix F. For every language, team names are sourced from the official rosters of the respective leagues, and their home cities are used as the localized set of cities. User names are sampled from the most frequent names in each country to ensure cultural plausibility and naturalness in the generated scenarios.

To further control cross-linguistic comparability, we manually translate all question templates, along with the function names and their descriptions that are exposed to the model during execution. An illustration of the interaction flow is provided in Figure 1. We enforce consistent constraints across languages and synchronize league schedules so that the distribution of games remains equivalent. This guarantees that differences in model performance during evaluation can be attributed to the agent's language-specific capabilities.

## 3.5 EVALUATION

A query is considered correct if the resulting environment state after the LLM execution matches the annotated state, that is, all expected games are booked and no unexpected games are booked. This means our evaluation is not dependent on LLMs. More details about the evaluation can be found at Appendix C.

Our main metric, **pass^k**, is adapted from code-generation benchmarks and estimates the probability that a model would succeed k independent attempts in the task. For each query $i$, let $c_i$ denote the

Table 3: pass^3 results for tested models in Ticket Bench. Results are displayed in each language covered in Ticket bench, and considering all questions in the benchmark.

| model | Pass^3 | | | | | | |
|---|---|---|---|---|---|---|---|
| | en | es | fr | it | de | pt | overall |
| GPT-5 | 0.92 | 0.93 | 0.92 | 0.92 | 0.92 | 0.87 | 0.91 |
| GPT-5 Mini | 0.91 | 0.91 | 0.89 | 0.90 | 0.89 | 0.86 | 0.89 |
| Qwen3-235B-A22B Yang et al. (2025) | 0.88 | 0.91 | 0.90 | 0.85 | 0.88 | 0.86 | 0.88 |
| GPT-5 Nano | 0.71 | 0.78 | 0.83 | 0.69 | 0.73 | 0.74 | 0.75 |
| GPT-OSS-120B Agarwal et al. (2025) | 0.73 | 0.76 | 0.72 | 0.70 | 0.73 | 0.67 | 0.72 |
| GPT-4.1 | 0.74 | 0.68 | 0.75 | 0.62 | 0.70 | 0.72 | 0.70 |
| Gemini-Pro 2.5 Comanici et al. (2025) | 0.85 | 0.54 | 0.76 | 0.48 | 0.59 | 0.6 | 0.63 |
| Gemini-Flash 2.5 Comanici et al. (2025) | 0.72 | 0.52 | 0.64 | 0.37 | 0.43 | 0.45 | 0.52 |
| Qwen3-32B Yang et al. (2025) | 0.40 | 0.55 | 0.51 | 0.56 | 0.55 | 0.56 | 0.52 |
| GPT-4.1 Mini | 0.48 | 0.59 | 0.49 | 0.52 | 0.54 | 0.48 | 0.52 |
| Qwen3-14B Yang et al. (2025) | 0.34 | 0.46 | 0.40 | 0.38 | 0.44 | 0.45 | 0.41 |
| Qwen2.5-72B-Instruct Qwen et al. (2025) | 0.25 | 0.34 | 0.47 | 0.30 | 0.42 | 0.48 | 0.38 |
| Qwen2.5-32B-Instruct Qwen et al. (2025) | 0.28 | 0.30 | 0.37 | 0.25 | 0.35 | 0.43 | 0.33 |
| Qwen3-30B-A3B Yang et al. (2025) | 0.24 | 0.34 | 0.36 | 0.31 | 0.38 | 0.35 | 0.33 |
| Sabia-3.1 Abonizio et al. (2024) | 0.30 | 0.24 | 0.27 | 0.21 | 0.29 | 0.30 | 0.27 |
| xLAM-2-32b-fc-r Prabhakar et al. (2025) | 0.22 | 0.24 | 0.21 | 0.27 | 0.30 | 0.31 | 0.26 |
| Qwen3-8B Yang et al. (2025) | 0.21 | 0.28 | 0.26 | 0.24 | 0.33 | 0.28 | 0.26 |
| GPT-OSS-20B Agarwal et al. (2025) | 0.29 | 0.27 | 0.21 | 0.14 | 0.29 | 0.31 | 0.25 |
| Qwen3-4B Yang et al. (2025) | 0.20 | 0.25 | 0.22 | 0.22 | 0.27 | 0.22 | 0.23 |
| GPT-4.1 Nano | 0.20 | 0.21 | 0.21 | 0.16 | 0.19 | 0.18 | 0.19 |
| Qwen2.5-14B-Instruct Qwen et al. (2025) | 0.16 | 0.18 | 0.22 | 0.12 | 0.11 | 0.25 | 0.17 |
| Qwen2.5-7B-Instruct Qwen et al. (2025) | 0.14 | 0.14 | 0.18 | 0.09 | 0.13 | 0.12 | 0.13 |
| Qwen2.5-3B-Instruct Qwen et al. (2025) | 0.10 | 0.12 | 0.12 | 0.08 | 0.11 | 0.10 | 0.11 |
| Llama-xLAM-2-8b-fc-r Prabhakar et al. (2025) | 0.08 | 0.08 | 0.12 | 0.10 | 0.14 | 0.06 | 0.10 |
| xLAM-2-3b-fc-r Prabhakar et al. (2025) | 0.03 | 0.11 | 0.07 | 0.05 | 0.06 | 0.05 | 0.06 |

number of correct executions across M runs. The empirical probability of success for each query is

$$p_i = \left(\frac{c_i}{M}\right)^k,$$

And the overall score is given by

$$\text{pass}\hat{3} = \frac{1}{N}\sum_{i=1}^{N} p_i.$$

This formulation rewards consistency across runs: queries that are solved correctly in multiple attempts contribute more than those solved only once. We compute metrics separately for each target language and in aggregate over the full multilingual dataset.

For this study, we set M=3 and K=3. We choose to run each model only 3 times for budget reasons, as some of the most expensive reasoning models can be notably expensive to run, GPT-5, for example, costs around $70 USD to run 3 times in all of Ticket Bench.

## 4 RESULTS

### 4.1 OVERALL PERFORMANCE

Table 3 reports the average pass^3 results across all tested models, broken down by language and overall performance.

The five best-performing systems—GPT-5 (0.91), GPT-5 Mini (0.89), Qwen3-235B Yang et al. (2025) (0.88), GPT-5 Nano (0.75), and GPT-OSS-120B Agarwal et al. (2025) (0.72)—all belong to

the category of reasoning models. These systems are designed to use more inference tokens to solve the task, at a higher computational cost, see Appendix D for further cost x performance analysis.

Outside the top models, accuracy starts to drop. GPT-4.1 (0.70 overall) remains a strong non-reasoning baseline, but its smaller variant GPT-4.1 Mini (0.52) falls behind, and GPT-4.1 Nano (0.19) shows a very lackluster performance. Qwen3-32B and Qwen3-14B show moderate performance (0.52 and 0.41), but the majority of Qwen2.5 models and smaller Qwen3 variants remain below 0.40 overall. This demonstrates that while instruction tuning improves usability, it does not provide the robustness required for complex multilingual function-calling.

An interesting trend emerges in the xLAM models Prabhakar et al. (2025), which were fine-tuned for function-calling based on BFCL and tau-bench tasks: their performance is consistently worse than their base Qwen2.5 models. For example, xLAM-2-32B-fc-r (0.26) underperforms Qwen2.5-32B-Instruct (0.33), and xLAM-2-3B-fc-r (0.06) falls behind Qwen2.5-3B-Instruct (0.11). This suggests that the specialized fine-tuning applied to xLAM may have improved capabilities in the target task but negatively impacted generalization across languages and tasks. These findings are consistent with the ones found by Acebench Chen et al. (2025).

For a more detailed analysis of the type of errors each model committed, see Appendix G.

## 4.2 SCALING TRENDS

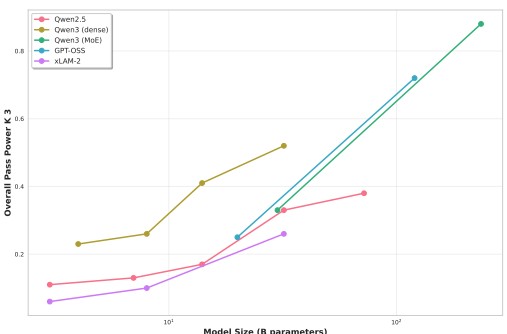
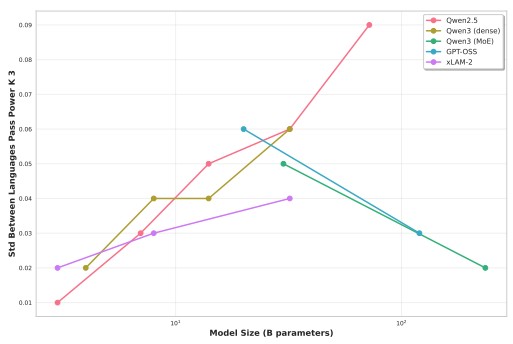

(a) Scaling Law of various LLMs families on Ticket-Bench.

(b) Standard deviation between different languages of various LLM families.

Figure 2: Scaling tendencies of open source models in Ticket Bench.

Figures 2a and 2b provide additional perspective by analyzing scaling behavior across model families.

Figure 2a illustrates the scaling behavior of different LLM families on Ticket Bench. The results follow a clear scaling law: as model size increases, accuracy steadily improves across most families. However, the slope of this improvement differs significantly. The Qwen3 (MoE) family shows the steepest growth, reaching competitive performance at large parameter counts. GPT-OSS models also follow a near-linear scaling trend, reinforcing the value of additional capacity. In contrast, the Qwen2.5 and xLAM-2 families scale more slowly, plateauing at considerably lower performance levels. This suggests that scaling alone is not sufficient; architectural and training choices (e.g., reasoning optimization) strongly mediate gains from larger parameter budgets.

Figure 2b reports the standard deviation of accuracy across languages for each family. Larger models generally exhibit greater cross-lingual consistency, as evidenced by the declining variance in GPT-OSS and Qwen3 (MoE) models at the highest scales. By contrast, Qwen2.5, Qwen3 (dense) and XLAM-2 models show increasing variance as parameters grow, indicating uneven improvements across languages.

## 4.3 CROSS-LINGUAL VARIATION

Figure 3 analyzes the relative performance of each model across languages by subtracting the model's overall average performance from its per-language score. Positive values indicate languages

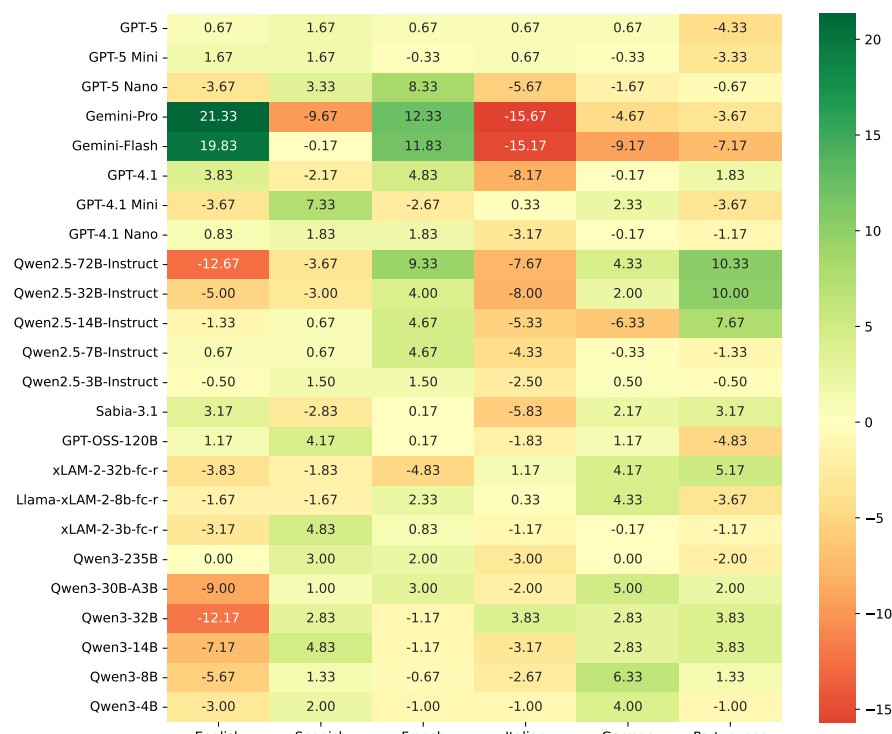

Figure 3: Heatmap showing the difference between each model per-language pass^3 performance and its own mean pass^3 performance among all languages. Models of the same family are displayed together.

where the model performs better than its own mean, while negative values highlight languages where the model performs worse than it's average performance.

**No "easy" or "hard" language across the board..** No single language consistently depresses scores across all systems. Instead, each language interacts differently with different families. For example, Qwen2.5-72B and Qwen2.5-32B achieve strong gains in Portuguese but show sharp deficits in English and Italian, while GPT-4.1 performs well in French yet struggles in Italian.

**Family-specific asymmetries**. Certain families show systematic biases. Qwen2.5 instruct models tend to favor French and Portuguese but lose accuracy in English and Italian. Qwen3 models also display a relative drop in English while improving in German and Spanish. Most surprisingly, both Gemini models display a disproportionate increase in English performance, with a secondary gain in French, but weaker results in all other languages. By contrast, the largest GPT-5 models maintain more balanced cross-lingual results, with the main exception being Portuguese, where deviations are more pronounced. These family-specific patterns are likely a reflection of the training data distribution used in each family.

**The best performing models are also more robust between languages..** The strongest models overall—GPT-5, GPT-5 Mini, and Qwen3-235B—are also the most consistent across languages. Their performance remains close to their own mean, indicating greater robustness. Nevertheless, even these models exhibit differences of at least five points between their best- and worst-performing languages, showing that cross-lingual performance remains an open challenge. Further analysis on cross-lingual performance can be found at Appendix E

Taken together, these findings show that multilingual function-calling remains uneven among most advanced models. While scale and reasoning can contribute to reducing variation, residual gaps across languages indicate that balanced and diverse multilingual training is still necessary.

## 5 CONCLUSION

In this work, we introduced Ticket-Bench, a benchmark designed to evaluate multilingual capabilities and variations in LLM agents. By simulating soccer ticket purchases across six major languages with localized teams, cities, and user profiles, Ticket-Bench provides a systematic and realistic framework for assessing LLM agent capabilities in multiple languages.

Our experiments reveal three central findings. First, reasoning-oriented models such as GPT-5 and Qwen3-235B show the most impressive performance. Second, scaling trends confirm that larger models generally achieve higher accuracy and more consistent results across languages, though families differ in their scaling efficiency. Third, cross-lingual variation remains a persistent challenge: no language is universally "easy" or "hard", and model families exhibit distinct asymmetries, underscoring the influence of training data distributions.

Ticket-Bench highlights both the progress of state-of-the-art reasoning models and their limitations. We hope that this benchmark will serve as a foundation for future research, encouraging the design of models and training regimes that are not only more powerful but also more equitable and reliable across the diverse linguistic and cultural contexts in which they will ultimately be deployed.

## 6 REPRODUCIBILITY STATEMENT

The code necessary to run the benchmark is available at `https://anonymous.4open.science/r/Ticket-Bench-5BC0`. The methodology and components of Ticket-Bench are explained in section 3. The complete list of templates used for English is shown in Appendix B. Further details about the evaluation and the models used can be found in Appendix C.

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

## A    USAGE OF LLMs

In the creation of this paper, LLMs were used to aid the writing process and to improve the flow of text. No LLMs were used during the idealization of the methodology or the elaboration of the results

## B    TEMPLATE LIST

This appendix shows the English version of all the templates used in Ticket Bench. Placeholders denoted between {} are dynamically filled when instantiating the templates.

1. Please buy a ticket for the next {user_team} game that I can afford.
2. Please buy a ticket for the next game of my team that I can afford.
3. Please buy a ticket for the next game of my team that I can afford, and that happens in the first semester of the year.

4. Please buy a ticket for the next game of my team that I can afford and that is not on a weekend.

5. Please buy a ticket for the cheapest game of my team that happens this year.

6. Please buy a ticket for the next game of my team that happens in {location}.

7. Please buy a ticket for the next game of my team that is against a team that scored more than 60 points in {year}.

8. Please buy a ticket for the next game of my team that happens in the second semester of the year and that takes place in the middle of the week (Tuesday, Wednesday, or Thursday).

9. Please buy a ticket for the most expensive game of my team that I can afford and that is not on a weekend.

10. Please buy a ticket for the cheapest game of my team that is in {location}.

11. Please buy a ticket for the next game of my team that happens in {location} and is against one of the top 8 teams of {year}.

12. Please buy a ticket for the cheapest game of my team that happens in the second semester of the year and that takes place in the middle of the week (Tuesday, Wednesday, or Thursday).

13. Please buy a ticket for the most expensive game of my team that I can afford and that is not on a weekend and that is in {location}.

14. Please buy a ticket for the cheapest game of my team that is in {location} and is against a team that scored more than 20 goals in {year}.

15. Please buy a ticket for the most expensive game of my team that I can afford and that happens in the second semester of the year, takes place in the middle of the week (Tuesday, Wednesday, or Thursday), and is in {location}.

16. Please buy a ticket for the cheapest game of my team that I can afford, that is not on a weekend, is in {location}, and is against a team that scored more than 20 goals in {year}.

17. Please buy a ticket for the most expensive game of my team that I can afford, that is in {location}, is against one of the top 3 teams of {year1} or {year2}, that is not on a weekend, and that happens in the second semester of the year.

## C EVALUATION DETAILS

The anonymized repository for Ticket-Bench is available in `https://anonymous.4open.science/r/Ticket-Bench-5BC0`. The repository contains the source code used to run evaluations, as well as all the data that comprises our benchmark, namely, user definition, schedule definitions, and the question sets.

This section provides more detailed information on our model evaluation process. All models were tested with a temperature of 0.7, except GPT-5 which enforces a temperature of 1 in its API. Models were allowed to generate up to 6 thousand tokens per round, and we enforced a limit of 20 interaction rounds (i.e a maximum of 20 sets of function calls), if the model reached such limit, the current task would be marked as wrong and the evaluation would continue.

Open source models up to 30B were used in a VM with 2A6000 GPUs, we used VLLM V.0.10.1 as the engine and interacted with models through the openai compatible endpoints exposed, all models were instanciated with the limit of at least 32k tokens and used the appropriate templates. Open source models larger than 30B were executed with the help of third-party providers that served the necessary models as paid endpoints. Finally, Maritaca AI and OpenaAI models were executed using the respective proprietary APIs.

One exception was the Gemini models; we were unable to run our experiments using the direct Gemini API provided by Google, due to limited usage quotas that made it infeasible to run the whole benchmark. As a workaround, we used Gemini through the proxy offered by Deepinfra.

Table 4 shows the used providers for each of the tested Models.

Table 4: Inference provider for all tested models in our research.

| Models | Inference provider |
|---|---|
| Open source models < 32B | VLLM 0.10.1, local GPUs |
| Qwen-2.5-72B-Instruct | Together AI API |
| Qwen3-235B-A22B | Together AI API |
| GPT-OSS-120B | DeepInfra API |
| Gemini Models | DeepInfra API |
| Sabia-3.1 | Maritaca AI API |
| GPT-4.1 models | OpenAI API |
| GPT-5 models | OpenAI API |

## D  COST ANALYSIS OF LLMS

This appendix provides an analysis of the trade-off between **cost** and **performance** of large language models (LLMs) based on Ticket-Bench results. Figure 4 presents the relationship between overall accuracy (measured with the pass[3] metric) and the total benchmark cost in USD (logarithmic scale). Costs combine prompt and completion tokens under published input/output rates. The x-axis is logarithmic, so small horizontal shifts can represent large price differences. All models ran the same workload with identical prompts and temperature; variation reflects token usage and pricing, not setup.

High-performing reasoning models, such as GPT-5 and Qwen3-235B, achieve the best results, with scores above 0.88, but they are also the most expensive to run. GPT-5 Mini stands out as a more efficient alternative, since it reaches nearly the same accuracy as GPT-5 while reducing costs significantly. This highlights that scaling down reasoning-oriented architectures can preserve robustness at a fraction of the price.

Mid-range models, including GPT-5 Nano and GPT-OSS-120B, deliver solid performance between 0.70 and 0.75 at moderate costs. These models represent a reasonable balance between reliability and affordability, making them good candidates for cost-sensitive deployments. Open-source options, particularly GPT-OSS, show a favorable cost-to-performance profile, proving competitive when compared to proprietary systems at similar scales.

Low-cost models, such as the Qwen2.5 Instruct family (3B–14B) and GPT-4.1 Nano, remain inexpensive but present limited accuracy, generally below 0.40. Their affordability is offset by low reliability, especially in multilingual function-calling tasks. Similarly, Gemini models (2.5 Flash and 2.5 Pro) are relatively expensive for their accuracy levels, which remain around 0.50–0.60.

In summary, reasoning-optimized models dominate performance but at a high financial cost. GPT-5 Mini and GPT-5 Nano offer a better cost-effectiveness trade-off within the GPT family, while GPT-OSS demonstrates that open-source alternatives can compete at a lower price point.

## E  CROSS-LINGUAL VARIATION ANALYSIS

This appendix presents an analysis of cross-lingual variation in model performance, as shown in Figure 5. The figure compares, for each model, its best-performing and worst-performing languages according to the pass[3] metric. The horizontal bars illustrate the performance gap between the two extremes.

The gap between the best and worst language differs by model scale and architecture. Larger reasoning-oriented systems such as GPT-5, GPT-5 Mini, and Qwen3-235B display smaller gaps, usually under 0.10 in absolute pass[3] score, which indicates greater robustness. Gemini models, in contrast, show a bias toward English and French with weaker results in other languages. Smaller or instruction-tuned models often show wide disparities exceeding 0.30, highlighting inconsistencies in multilingual handling.

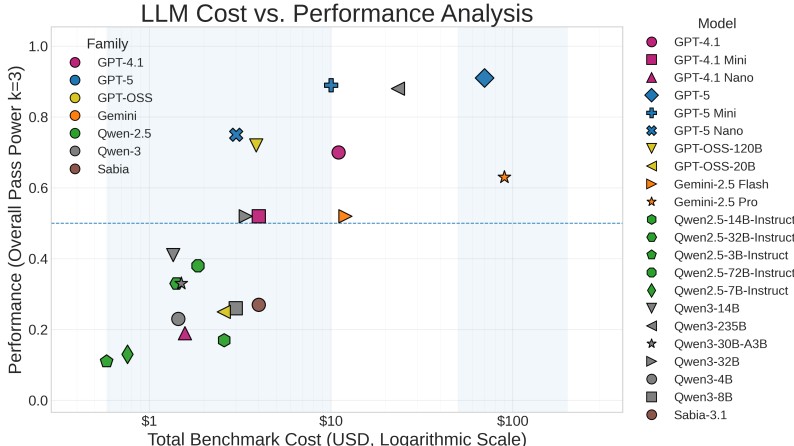

Figure 4: Cost vs. performance analysis of LLMs on Ticket-Bench. The x-axis shows the total benchmark cost in USD (log scale), while the y-axis shows overall performance using the pass[3] metric.

Finally, the figure reinforces the observation that multilingual robustness remains a challenge even for advanced systems. While scaling improves overall consistency, family-specific asymmetries tied to training data distributions persist.

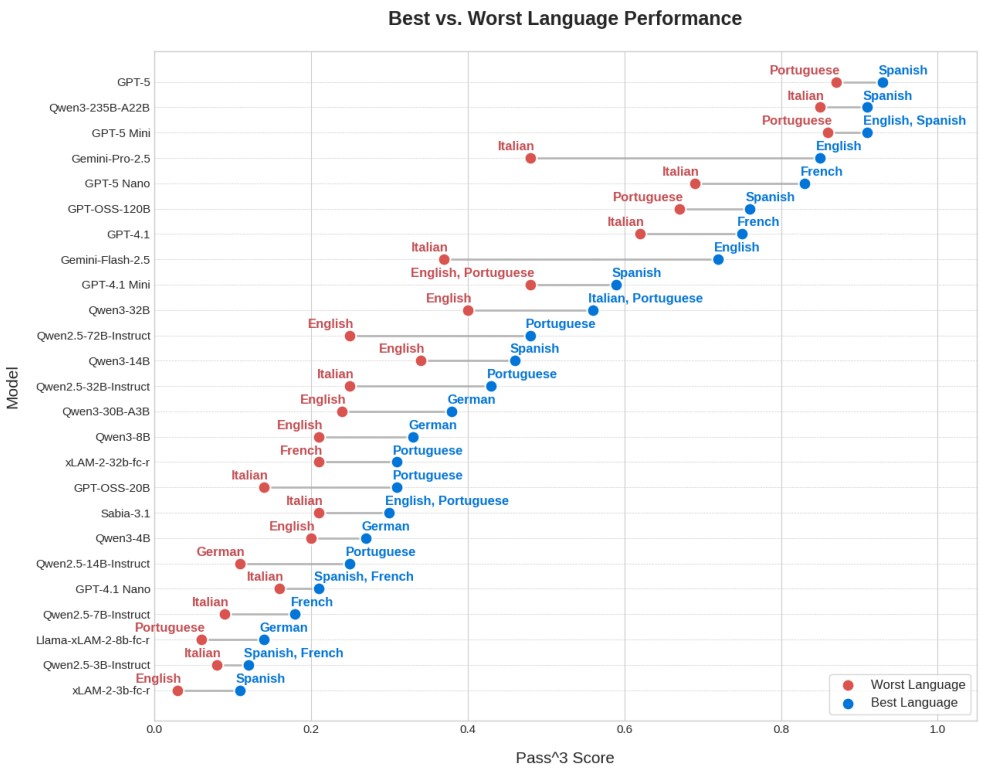

Figure 5: Best vs. worst language performance for each model on Ticket-Bench. Red markers indicate the lowest-scoring language per model, blue markers the highest-scoring language. Horizontal bars show the performance gap.

## F  LEAGUE/LANGUAGE SELECTION CRITERIA

To represent the most relevant football competitions across different languages, we selected the Premier League (English), Ligue 1 (French), Serie A (Italian), Brasileirão Série A (Portuguese), Bundesliga (German), and La Liga (Spanish). These leagues were chosen for their combination of competitive strength in recent years FIFA (2025); of Football History & Statistics (2023), financial relevance SportingPedia (2025), international influence Wright (2022), and historically successful national teamsOlympics (2024). They consistently perform at the highest levels in continental and global competitions and represent the leading football cultures in their respective languages and regions.

## G  ERROR ANALYSIS

This section aims to examine the errors made by the models during the execution of the benchmark. Given the specific patterns of questions, it is possible to programmatically detect these errors and determine both their nature and the stage of agentic reasoning in which they occurred. To address this, we have implemented several mandatory checklists that a model must satisfy to fully accomplish the task. Our analyses will be based on these checklists that are presented below for each of the questions in the appendix B.

**Common Checklists Applied to All Questions:**

1. **User Info Is the First Tool Call**, since all questions require specific knowledge about the user, such as the user team or the money balance, this checklist verifies whether the first tool call made by the model was to read the user's information.

2. **Listed Games**: To solve the task, models must list the available games at least once.

3. **Bought Correct Number of Games**, did the models buy exactly one game?

4. **Bought the Correct Game**, models are expected to buy the correct (i.e., expected) game.

5. **User Can Afford**, did the model attempt to buy a game the user could not afford?

**Specific Checklists** In the following, we present the specific checklists applied only to the questions where it is required.

1. **Correct location**: Whether the models bought a game in the correct location.

2. **Correct opponent score**: This verifies whether the model attempted to buy only games where the required opponent score was satisfied, such as being in the top 8 on the leaderboard or having more than 60 points.

3. **Correct price choice**: Whether the model succeeded in buying a game that met the price restrictions, such as being the cheapest or most expensive game among the possible options (considering affordability, location, etc).

4. **Correct period**: Whether the models bought a game scheduled in the correct time period, such as on the weekend, specific days of the week, or within a particular semester.

5. **Used leaderboard**: Some questions require inspecting the leaderboard. Did the models use it at least once?

**Heatmaps**

We present the heatmaps of the analysis, showing the percentage of success across all checklists. These visualizations provide a comparative view of model behaviors for each language.

Figure 6 illustrates the results for German, followed by the English case in Figure 7, Spanish (Figure 8), French (Figure 9), Italian (Figure 10), and Portuguese (Figure 11). Together, these six heatmaps provide a comprehensive multilingual perspective on checklist-level variation. It is important to note that each checklist is independent of whether the model bought the correct game; that is, if the model purchased a game within the correct time period, the checklist item is marked as True, regardless of whether it was the correct game.

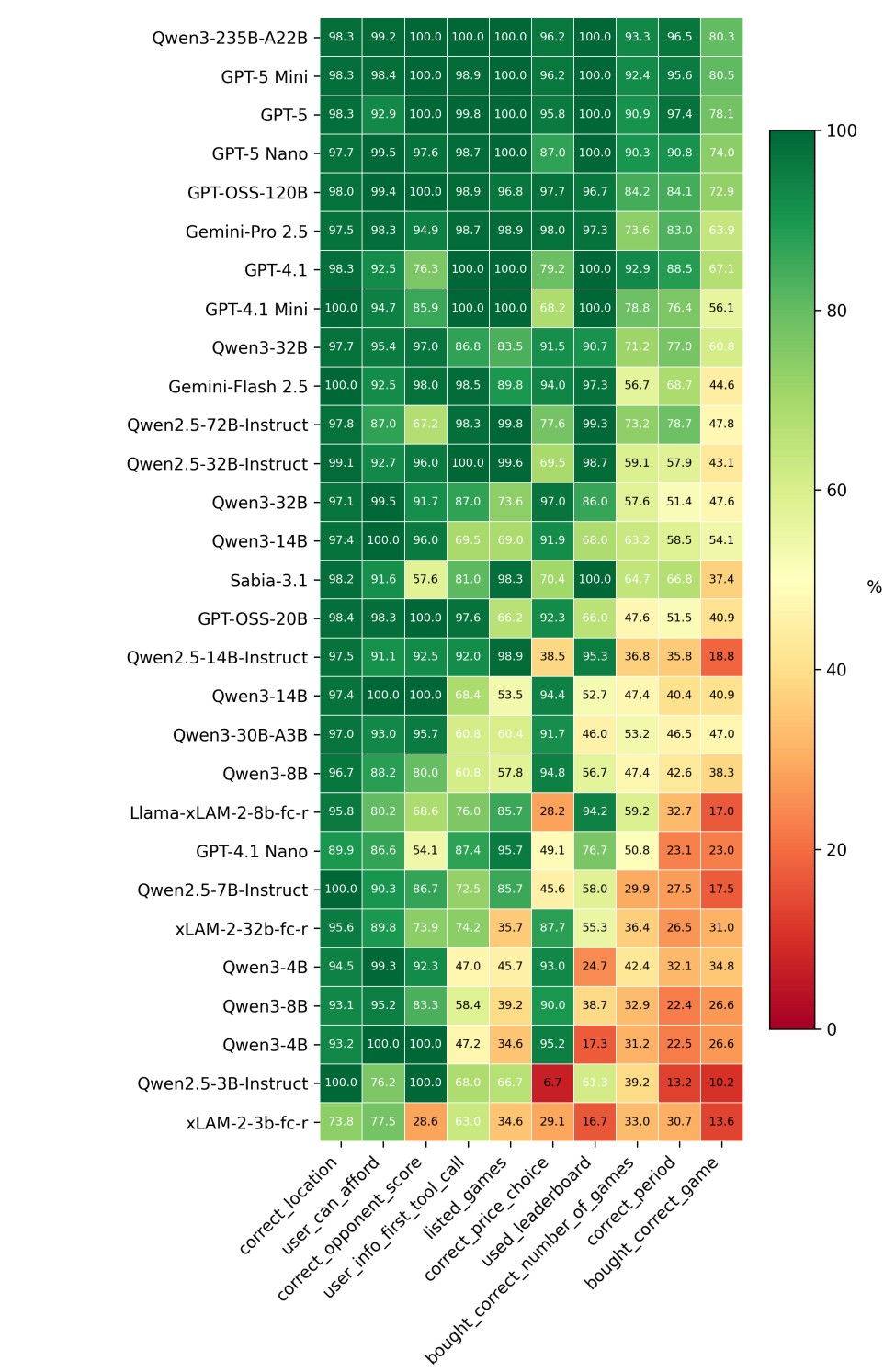

Figure 6: German heatmap of success rates for each error reason checklist.

Other limitations might also apply. For example, smaller models often fail to buy any game at all in a significant number of matches. In such cases, certain checklists cannot be computed and are therefore set to None, primarily because there is no game to evaluate.

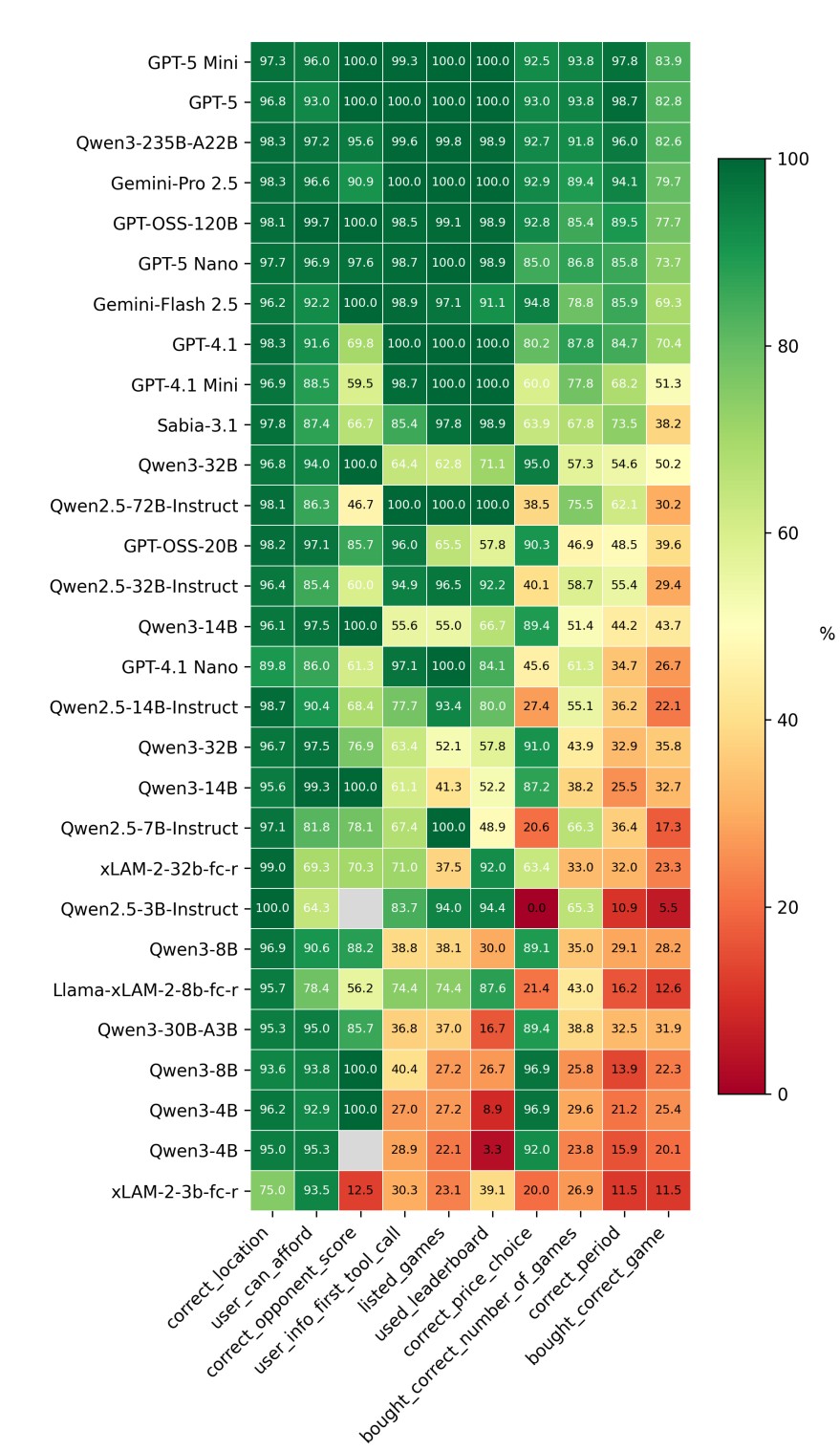

Figure 7: English heatmap of success rates for each error reason checklist.

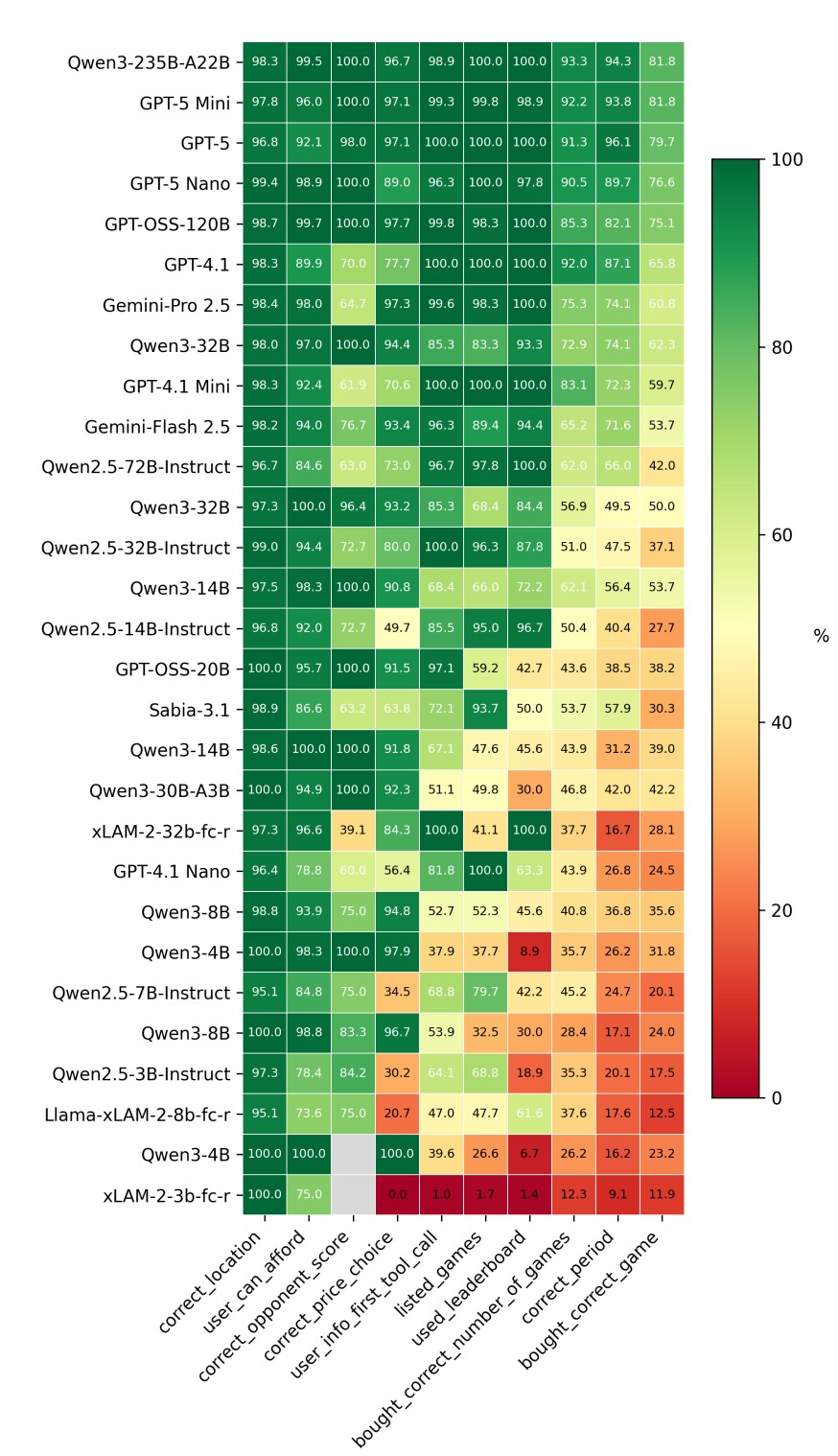

Figure 8: Spanish heatmap of success rates for each error reason checklist.

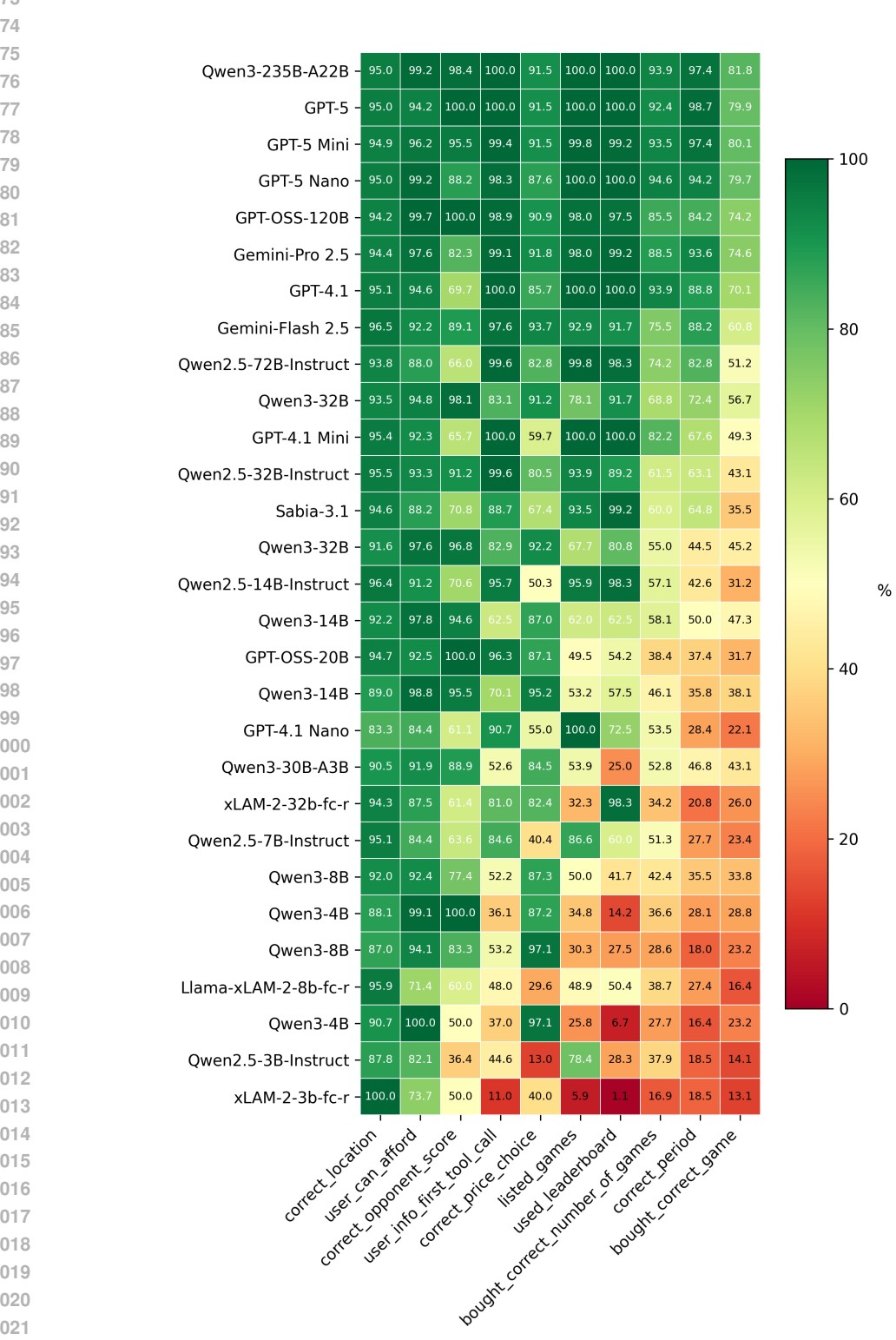

Figure 9: French heatmap of success rates for each error reason checklist.

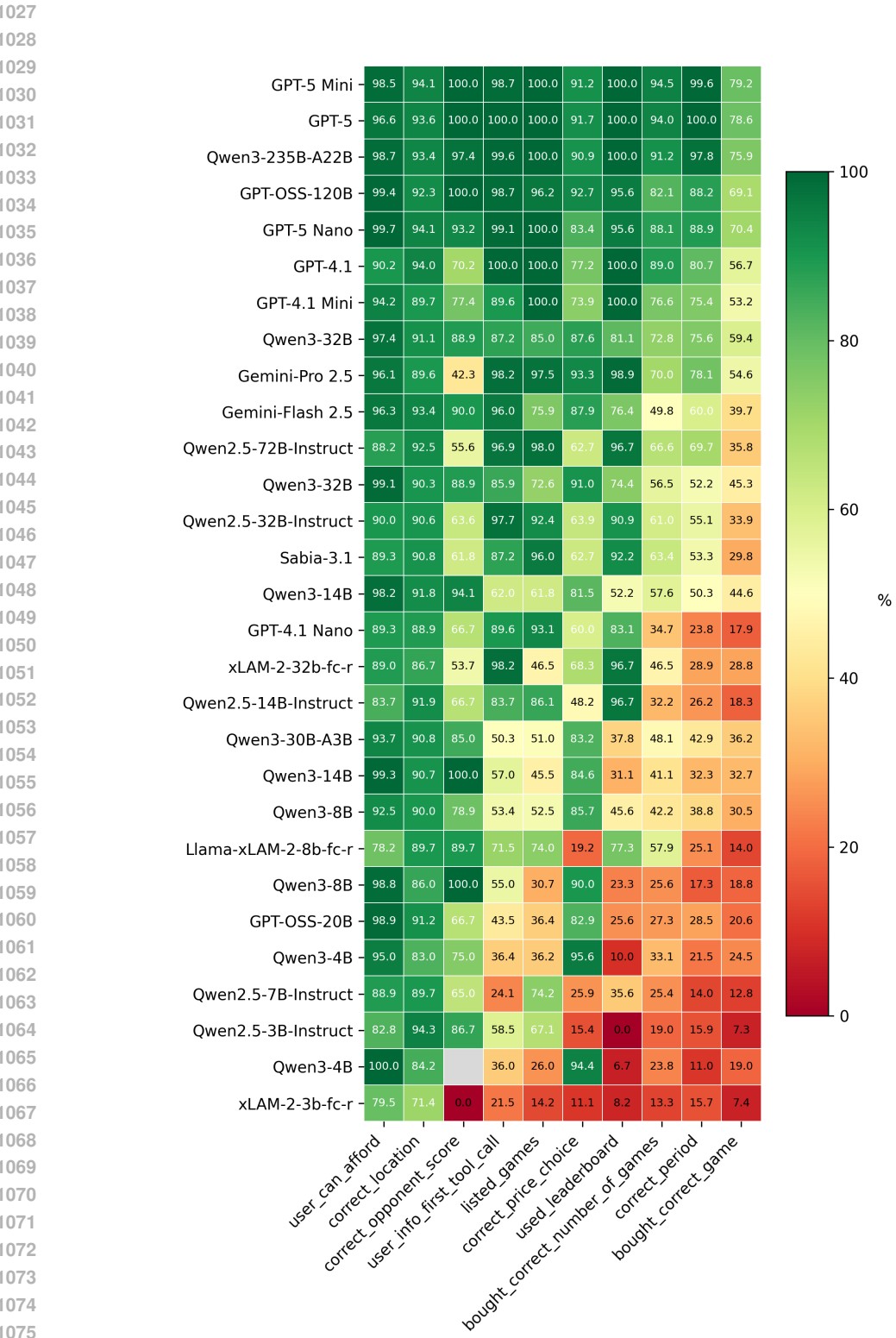

Figure 10: Italian heatmap of success rates for each error reason checklist.

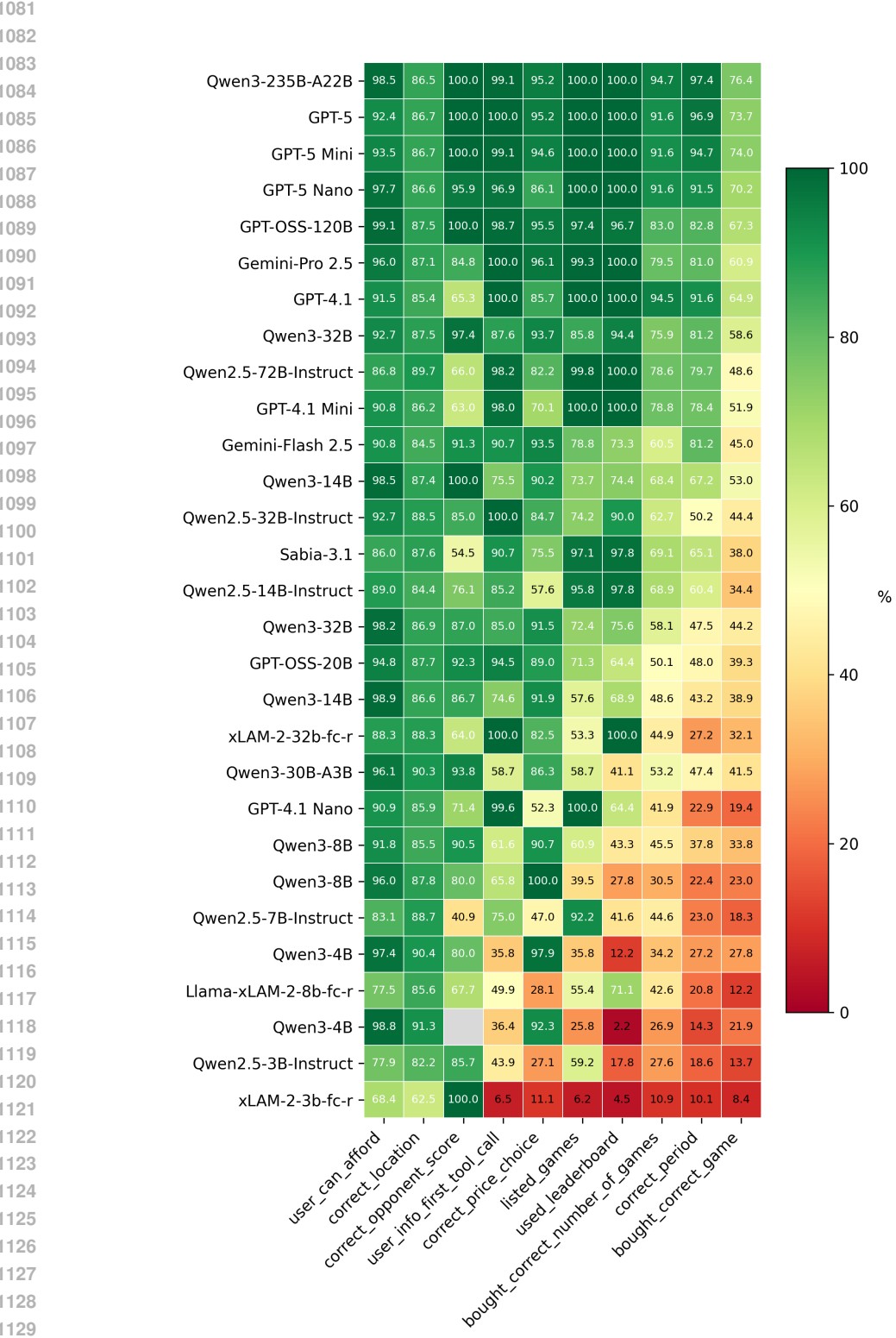

Figure 11: Portuguese heatmap of success rates for each error reason checklist.

