# OpenReview forum: "Ticket-Bench: A Kickoff for Multilingual and Regionalized Agent Evaluation"
_ICLR.cc/2026/Conference — Submitted to ICLR 2026_

### Official Review · Reviewer_uUwC · 2025-10-30

**Soundness:** 2
**Presentation:** 2
**Contribution:** 2
**Rating:** 4
**Confidence:** 3

**Summary:**

The paper highlights the need for multilingual, culturally aware benchmarks for evaluating function calling. The authors introduce Ticket Bench to address this issue. Ticket-Bench features tasks in six major languages– Portuguese, English, Spanish, German, Italian, and French to evaluate LLM agent capabilities in ticket-purchasing scenarios. They build the benchmark by simulating user profiles, localized teams and cities. They compare the performance of various models across different languages on the benchmark and show that GPT-5 and Qwen3-235B are most robust.

**Strengths:**

1. The paper highlights the need for agentic evaluation frameworks in multilingual settings to evaluate function calling. The paper introduces a benchmark TicketBench, which simulates a real world scenario i.e purchasing soccer tickets.

2. The ideas in the paper are presented clearly along with results across multiple model families in Table 3 and cross-lingual variation in Figure 3.

**Weaknesses:**

While the benchmark is trying to address a real world scenario, the usefulness of the benchmark seems to be limited. The benchmark uses 17 question templates to represent distinct ticket purchasing scenarios. This limits the diversity of the instructions given to the agent and do not adequately represent the diversity of real world queries. Similarly, the agent has access to only five callable functions in the entire benchmark.

The paper could provide additional analysis of the rate of success for each function invocation across models.

**Questions:**

1. How are the question templates translated into other languages? Are these manually verified by the authors to ensure coherence?

---

> ### Author Response · Authors · 2025-12-03
>
> First, we would like to sincerely thank the reviewer for the thoughtful and constructive comments. The feedback is extremely valuable and helps us better assess the strengths and limitations of our work, as well as clarify aspects that may not have been sufficiently explicit in the paper.
>
> About the weaknesses listed by the reviewer, we provide our detailed responses below.
>
> W1- The reviewer says “While the benchmark is trying to address a real world scenario, the usefulness of the benchmark seems to be limited.”. We disagree about the claim that the benchmark is not useful, we show a notable gap in performance in LLMs for different languages and regions, we do not know of a similar work that showed such points in an agentic task. While the expansion of domains is certainly a promising future direction, we believe that Ticket-Bench already demonstrates an important signal of notable performance gaps in its current version.
>
> W2- The reviewer also expresses doubts about the number of templates used, while we agree that more robust ways of improving variety and incorporating other regional factors can enhance the benchmark, we believe they are not fundamental, as with the current level of regionalization we do perform, we already see significant gaps between models' performances in different languages.
>
> About the questions
>
> Q1- Yes, the questions were manually translated and checked.

---

### Official Review · Reviewer_fWJZ · 2025-10-30

**Soundness:** 3
**Presentation:** 2
**Contribution:** 2
**Rating:** 4
**Confidence:** 4

**Summary:**

The paper introduces Ticket-Bench, a benchmark created to assess LLMs in task-oriented, realistic, multilingual environments. It focuses on function-calling in the domain of soccer ticket purchases across six languages (Portuguese, English, Spanish, German, Italian, and French). Each language has its own culturally appropriate teams, cities, and user profiles. Unlike previous monolingual or naively translated benchmarks, Ticket-Bench captures both linguistic and cultural diversity, providing a structured environment that evaluates models based on the correctness of their final actions. To test the ability to handle different logical constraints, like affordability, location, day of the week, or previous team performance, the authors defined 17 different ticket purchasing templates.

**Strengths:**

- Multilingual and culturally grounded evaluation.
- Carefully designed methodology that supports LLM-free evaluation.
- Benchmark covers six fully localized languages.

**Weaknesses:**

- Benchmark focuses solely on soccer ticket purchases.
- Study relies exclusively on programmatic evaluation without inspecting model reasoning quality.
- Benchmark excludes low-resource languages.
- Manuscript contains multiple typos and inconsistent capitalization (e.g., “OpenaAI” instead of “OpenAI”; double periods like “..”).

**Questions:**

- Do you plan to expand Ticket-Bench to include non-European and low-resource languages in future releases?
- Did you check whether differences in prompt length across languages affect accuracy?
- Could translated function names cause bias, and did you test using shared English names across languages?
- How sensitive were the results to the verbosity of tool descriptions and examples in the function schema?

---

> ### Author Response · Authors · 2025-12-03
>
> First, we would like to sincerely thank the reviewer for the thoughtful and constructive comments. The feedback is extremely valuable and helps us better assess the strengths and limitations of our work, as well as clarify aspects that may not have been sufficiently explicit in the paper.
>
> About the weaknesses listed by the reviewer, we provide our detailed responses below.
>
> W1- Yes, our benchmark focus on buying tickets for soccer games, we believe it was a good starting point for the analysis presented, the major point of the study is showing the LLM performance gaps in a realistic task, while expanding domains is an interesting future work, we don’t believe it hurts our results.
>
> W2- Yes our study focuses on the programmatic evaluation of the problems, and does not evaluate the model reasoning text, but we do have an error analysis that studies why the model gets to a wrong conclusion in appendix G. We believe that the programmatic only evaluation is not necessarily a weakness, we are most interested if the model can perform the tasks we are proposing, and this way we  do not have uncertainties such as a model receiving higher scores such because it speaks in a certain way.
>
> W3- Our language selection focuses on languages and regions that would be more likely to use an agent for buying tickets for soccer games, therefore getting closer to a real application. Expansion to other languages is interesting but we already cover languages that are not often covered for benchmarks for agentic systems, such as Portuguese.
>
> W4- We thank the reviewer for pointing this out. These will be carefully corrected.
>
>
>
> Now about the presented questions:
> Q1- Yes, alongside doing other domains, adding more languages is one of the possible expansions we are considering.
> Q2- We did not include such analysis, but the difference between prompt sizes is very small, so it is very unlikely to be a deciding factor.
> Q3- Possibly, but we believe that this kind of effect is very model dependent. In the paper we note for example how models of a same family show the same tendencies of strong/weak languages.
> Q4- Due to the cost of running so many models on Ticket-Bench, we refrained from performing too many variations on the tool descriptions.

---

### Official Review · Reviewer_ZSe4 · 2025-11-01

**Soundness:** 2
**Presentation:** 3
**Contribution:** 2
**Rating:** 2
**Confidence:** 4

**Summary:**

Ticket-Bench is a novel evaluation benchmark for LLM Agents, focusing on addressing the prevalent problems of "English-centric" evaluation and a "lack of cultural authenticity." The benchmark simulates a "soccer ticket purchasing" task scenario. It covers six major languages (Portuguese, English, Spanish, German, Italian, and French). Unlike previous datasets that rely merely on translation, Ticket-Bench ensures "cultural authenticity" by using real, localized entities corresponding to each country in different language versions (such as team names, city names, and common user names).

**Strengths:**

The work involves a substantial engineering effort.

The writing is fluent.

**Weaknesses:**

In terms of language: Manually translating to generate queries is a traditional method, which leads to an evaluation environment that is overly "clean" and standardized. It fails to reflect the complexity of real-world user queries, such as colloquialisms, ambiguity, grammatical errors, or expressions with mixed intent. Furthermore, the six languages were all selected based on their relevance to "soccer leagues," which results in the selected languages being primarily concentrated in Europe and South America.

In terms of regionalization: The benchmark largely equates "cultural authenticity" with "entity localization" (i.e., replacing team names, city names, and personal names). This approach may fail to capture deeper cultural differences, such as local customs, habits, and cultural context. It is questionable whether the soccer ticket domain can generalize to other domains that require regionalized knowledge.

**Questions:**

Does the benchmark include tests for complex agentic behaviors?

---

> ### Author Response · Authors · 2025-12-03
>
> First, we would like to sincerely thank the reviewer for the thoughtful and constructive comments. The feedback is extremely valuable and helps us better assess the strengths and limitations of our work, as well as clarify aspects that may not have been sufficiently explicit in the paper.
>
> About the weaknesses listed by the reviewer, we provide our detailed responses below.
>
> W1- We agree that simply translating the templates does not necessarily collect colloquialisms, idioms, or culturally specific expressions and incorporating such things is an interesting evolution. However, we do not believe that the absence of these aspects undermines our conclusions. Even with direct translations and the addition of region-specific elements such as local soccer teams, we already observe notable performance gaps across the evaluated LLMs. This indicates that the benchmark is capable of revealing meaningful cross-regional differences even without deeper cultural adaptation.
>
> Regarding the selection, it is true that our criteria for selecting languages lead us to primarily concentrate in Europe and South America. However, our selection also leads us to study the agent capacity in scenarios where it would be most used, leading to a more meaningful evaluation. We also would like to point out that the number of benchmarks covering South American countries is minimal, and we believe Ticket-Bench helps cover this gap.
>
> W2- The reviewer claims “It is questionable whether the soccer ticket domain can generalize to other domains”. We believe that this is an interesting discussion, but we believe that should not deviate from the contribution Ticket-Bench already presents, we show that current LLMs show a notable gap in performance under simple regionalization and language shift, future work can further expand this and try to understand if such gaps appear in the same way in other domains.
>
>
> Q1- The reviewer asked “Does the benchmark include tests for complex agentic behaviors?”. What is a ‘complex agentic behavior’ is not defined, but we can point out that the harder question templates involve multiple constraints, for the model to perform well in them it would need to perform up to 7 tool calls with sequentially dependent outputs. We believe this could qualify as a complex agentic task.

---

### Official Review · Reviewer_1NUn · 2025-11-07

**Soundness:** 2
**Presentation:** 1
**Contribution:** 2
**Rating:** 2
**Confidence:** 4

**Summary:**

This paper introduces Ticket-Bench, a new benchmark for evaluating LLM agents' function-calling capabilities in multilingual and regionalized scenarios. The authors argue that existing benchmarks are overly English-centric. To fix this, Ticket-Bench simulates a soccer ticket-purchasing task across six languages, crucially binding each language to realistic, local entities (e.g., Brazilian league teams for Portuguese, German league teams for German) rather than just translating text.

The authors use a LLM-free evaluation that checks the final task state and test a wide range of LLMs. They find that reasoning-oriented models (like GPT-5 and Qwen3-235B) perform best and that performance generally scales with model size. They also find significant cross-lingual disparities, with different model families showing distinct performance patterns across the six languages, likely reflecting their training data.

**Strengths:**

1. Sound Motivation: The motivation to create a "regionalized" agent evaluation benchmark is practical. It addresses a gap in existing evaluations which are often English-centric and overlook local context.
2. Broad Model Coverage: The benchmark is evaluated on a wide and abundant range of LLMs, including both prominent commercial and open-source models of various scales.
3. Robust Evaluation Metric: The use of pass^k is a sound choice, offering a more stable and reliable measure of model consistency compared to a single-pass metric.

**Weaknesses:**

1. Limited Scope of Agent Capabilities Evaluated: The paper presents a benchmark for agent evaluation, but its scope is limited, focusing primarily on multi-step tool use rather than higher-level agentic capabilities. Specifically, the benchmark does not evaluate the critical autonomous task planning ability of agents. Instead, its queries are structured (17 templates), which pre-defines the task's logical decomposition. Also, the evaluation environment assumes successful API calls. A robust agent, however, must be able to handle API failures or dynamic environmental changes. The benchmark does not assess this crucial capacity for error detection and autonomous refinement.

2. Disconnect Between Benchmark Design and Experimental Analysis. A core weakness of this paper is the significant disconnect between its novelty of "regionalized" design (Section 3.4) and its subsequent "linguistic" analysis (Section 4.3). Specifically, the methodology commendably creates "regionalized" data by binding languages (e.g., pt) to specific local entities (e.g., Brazilian "Brasileirão" teams). However, the results analysis (Section 4.3) regresses to a simple "cross-lingual" comparison, only analyzing performance differences across the six languages. This fails to answer the paper's own core research question. When a model fails on a German (de) task, the analysis does not provide insight into whether the failure is due to a lack of linguistic understanding (German grammar) or a lack of regional knowledge (German "Bundesliga" facts). This leaves the main "regionalization" contribution feeling hollow at the analysis stage.

3. Superficial Analysis and Misplaced Error Analysis. The experimental analysis is superficial, relying entirely on top-level quantitative metrics (e.g., pass^3) without providing deeper insights. It has two shortcomings:
- Lack of Qualitative Case Studies: The main paper includes no qualitative analysis or case studies. A single example of a failed API call sequence would have been invaluable for understanding the root cause of model failures and for supporting the "linguistic vs. regional" argument.
- Misplaced Error Analysis: A more detailed, programmatic error analysis (e.g., "did the model call Get_Leaderboard?") does exist, but it is relegated to Appendix G. This granular analysis is important, and I would suggest it to be in the main text to explain why different model families (e.g., Gemini vs. Qwen) show different top-level performance.

4. Poor Presentation and Lack of Rigor. The paper suffers from poor presentation, which is not up to the standard of a top-tier conference. For example, there are rendering errors and notation missing issues: The metric definition in Section 3.5 (line 307) for pass^3 has a clear rendering error in the PDF, with the caret ^ overlapping the number 3. Also, the symbol N in the formula is not defined.

**Questions:**

1. The term "agent evaluation" seems overstated, I would suggest the authors to justify this scope or re-frame the paper as a "tool-use" benchmark.
2. The current results don't show if models fail due to language or regional knowledge. Can authors provide some analysis that disentangles   these two factors?
3. The paper notes that xLAM models, fine-tuned for function calling, performed worse than base models. This is counter-intuitive. What is the authors' hypothesis for this?

---

> ### Author Response · Authors · 2025-12-03
>
> First, we would like to sincerely thank the reviewer for the thoughtful and constructive comments. The feedback is extremely valuable and helps us better assess the strengths and limitations of our work, as well as clarify aspects that may not have been sufficiently explicit in the paper.
>
> About the weaknesses listed by the reviewer, we provide our detailed responses below.
>
> W1- The reviewer “The paper presents a benchmark for agent evaluation, but its scope is limited, focusing primarily on multi-step tool use rather than higher-level agentic capabilities. Specifically, the benchmark does not evaluate the critical autonomous task planning ability of agents”. We respectfully disagree., Although the Ticket-Bench uses templated questions, the agent being tested still needs to analyze and decide a coherent sequence of tools to achieve the objective.
> For instance, several questions require the agent to fetch leaderboard data from previous years (e.g., checking the top 3 teams of 2022) to infer which games remain eligible to be booked later. This type of reasoning, gathering intermediate information, interpreting it, and planning subsequent steps, constitutes precisely the kind of autonomous multi-step decision-making that characterizes agentic behavior, and Ticket-Bench explicitly tests for it.
>
> W2- The reviewer says “The methodology commendably creates "regionalized" data by binding languages (e.g., pt) to specific local entities (e.g., Brazilian "Brasileirão" teams). However, the results analysis (Section 4.3) regresses to a simple "cross-lingual" comparison, only analyzing performance differences across the six languages.” We appreciate this observation and agree that, based on our current experiments, we cannot fully disentangle whether the observed performance gap is driven primarily by regionalization or by language shift.
> While we recognize that isolating these factors is an interesting research direction, our intention in Section 4.3 was to highlight the existence of a notable gap in the realistic setting where both regionalization and language are simultaneously at play, a scenario that naturally occurs when deploying agents in specific regions (e.g., an agent operating in Brazil will interact in Portuguese and handle Brazilian teams). To isolate the two effects, we would need to rerun the entire benchmark using the same regional entities while varying all languages, which would require effectively doubling the number of experiments. As disclosed in the paper, running the benchmark with GPT-5 alone costs approximately 70 USD; repeating the full set of experiments across all models would add roughly 600 USD to the total cost, which exceeds our available budget.
>
> W3- As the reviewer points out, more nuanced error analysis is present in our appendix, this was an editing choice made due to the limited page number of 9 that ICLR establishes. But we would like to say that given more space ( like the 10 pages of the accepted publications) we would move this appendix up to the main text.
>
> W4- We respectfully disagree with the reviewer's overall assessment regarding the "poor presentation" of the paper. We fully acknowledge the specific issues raised: the formula rendering error and the missing explanation of the variable N. These concrete points are helpful and will be corrected. However, beyond these two issues, the reviewer does not provide additional details about what aspects of the presentation are considered poor. As a result, it is unclear whether the concern refers only to these minor points or to broader aspects of the paper.
> We assure the reviewer that the identified errors will be fixed in the camera-ready version, as they are minor points. If additional presentation issues were intended, we would be grateful for further clarification, as we are committed to improving the paper.
>
> Now about the presented questions
> Q1- We refer back to the discussion provided in Weakness 1. As explained there, we believe that Ticket-Bench does evaluate meaningful agentic capabilities of LLMs. Given the reviewer’s impression, we agree that the paper would benefit from a clearer and more explicit discussion of how agentic behaviors are captured and evaluated.
>
> Q2- Discussed on Weakness 2.
>
> Q3- In the final of section 4.1 in our paper, we say “This suggests that the specialized fine-tuning applied to xLAM may have improved capabilities in the target task but negatively impacted generalization across languages and tasks. These findings are consistent with the ones found by Acebench Chen et al. (2025)”. We believe that the xLAM train was too specific and did not lead to a model generalization, which makes xLAM perform worse in our benchmark.

---

### Comment · Area_Chair_Hy2S · 2025-11-28
**please respond to reviewer concerns promptly**

Dear Authors,

A quick reminder that the ICLR 2026 rebuttal window will close in less than a week. To ensure a fair and thorough evaluation, we encourage you to address the reviewers’ concerns as soon as possible.

Thank you for your prompt attention and for helping us keep the process efficient.

AC

---

### Meta-Review · Area_Chair_Z2Rp · 2025-12-27

**Summary:**

This paper introduces Ticket-Bench, a new benchmark for evaluating LLM agents' function-calling capabilities in multilingual and regionalized scenarios. The authors argue that existing benchmarks are overly English-centric. To fix this, Ticket-Bench simulates a soccer ticket-purchasing task across six languages, crucially binding each language to realistic, local entities (e.g., Brazilian league teams for Portuguese, German league teams for German) rather than just translating text.

The authors use a LLM-free evaluation that checks the final task state and test a wide range of LLMs. They find that reasoning-oriented models (like GPT-5 and Qwen3-235B) perform best and that performance generally scales with model size. They also find significant cross-lingual disparities, with different model families showing distinct performance patterns across the six languages, likely reflecting their training data.

**Reviewer Concerns:**

Advantages:
1. Sound Motivation: The motivation to create a "regionalized" agent evaluation benchmark is practical. It addresses a gap in existing evaluations which are often English-centric and overlook local context.
2. Solid evaluation: there is multilingual and culturally grounded evaluation including broad model coverage and robust evaluation metric, and the benchmark covers six fully localized languages.

Disadvantages: d
1. There is limited scope of agent capabilities which were evaluated.
2. The benchmark focuses solely on soccer ticket purchases, which largely limits the usefulness of the benchmark in real world applications.
3. The analysis can be much improved. There is disconnect between benchmark design and experimental analysis, and lack of qualitative case studies.
4. The benchmark can be improved in terms of including low-resource languages and capturing more realistic language uses and cultural differences.

**Reviewer Scores:**

reviewers gave scores like 2,2,4,4 pretty consistently among them.

---

### Decision · Program_Chairs · 2026-01-26

Reject